# Microstructure Formation and Carbon Partitioning with Austenite Decomposition during Isothermal Heating Process in Fe-Si-Mn-C Steel Monitored by In Situ Time-of-Flight Neutron Diffraction

Yusuke Onuki [1],*, Kazuki Umemura [2], Kazuki Fujiwara [3], Yasuaki Tanaka [4], Toshiro Tomida [1], Kaori Kawano [4] and Shigeo Sato [2]

[1]  Frontier Research Center for Applied Atomic Sciences, Ibaraki University, 162-1 Shirakata, Tokai 319-1106, Ibaraki, Japan; toshiro.tomida.tomida_tsr@vc.ibaraki.ac.jp

[2]  Graduate School of Science and Engineering, Ibaraki University, 4-12-1 Nakanarusawa, Hitachi 316-8511, Ibaraki, Japan; 20nm911l@vc.ibaraki.ac.jp (K.U.); shigeo.sato.ar@vc.ibaraki.ac.jp (S.S.)

[3]  Nippon Steel Research Institute Corporation, 1-8 Fuso-Cho, Amagasaki 660-0891, Hyogo, Japan; fujiwara.4h8.kazuki@jp.nipponsteel.com

[4]  Advanced Technology Research Labs., R&D Laboratories, Nippon Steel Corporation, 20-1 Shintomi, Futtsu 293-8511, Chiba, Japan; tanaka.km5.yasuaki@jp.nipponsteel.com (Y.T.); kawano.xp4.kaori@jp.nipponsteel.com (K.K.)

*  Correspondence: yonuki@mail.dendai.ac.jp; Tel.: +81-29-352-3234

**Abstract:** Retained austenite is a key feature used to realize the transformation-induced plasticity in bainitic high strength steels. In this study, the authors focused on the formation of metastable austenite in Fe-0.61C-1.9Si-0.98Mn (mass%) during isothermal heating processes using in situ neutron diffraction techniques. Quantitative discussion of carbon partitioning processes is enabled by applying an in situ phase fraction analysis considering crystallographic textures, in addition to the carbon concentration estimation based on the lattice parameter of austenite. The carbon partitioning behavior is inhomogeneous, resulting in a bimodal carbon concentration distribution in austenite. The carbon enriched, high carbon austenite is stable during isothermal heating at 673 K and is retained even after cooling to room temperature. The remainder is low carbon austenite, which is gradually consumed by bainite transformation. Above 723 K, the high carbon austenite also decomposes to ferrite and cementite due to the fast diffusion of Si. Conversely, below 623 K, cementite is stabilized even without the diffusion of Si. These cementite formation mechanisms prevent the formation and retention of high carbon austenite. The inhomogeneous carbon distribution and cementite formation must be carefully considered to precisely predict the microstructure formation in Si-added bainitic steels.

**Keywords:** TRIP steel; bainite transformation; neutron diffraction; Rietveld texture analysis; iMATERIA

## 1. Introduction

Transformation induced plasticity (TRIP) steels are now widely used in the automotive industry [1]. The TRIP effect in Fe-Si-Mn-C is achieved by deformation-induced transformation of retained austenite. Therefore, the amount, shape, and chemical stability of retained austenite directly affect the mechanical properties of the steel.

The retained austenite is formed due to the bainite transformation during the austempering, which is an isothermal heating process after quenching from the austenizing temperature. The carbon expelled from bainitic ferrite increases the carbon concentration in austenite, resulting in the chemical stabilization. In Si-added steel, the carbon-enriched austenite can endure decomposition during isothermal heating due to the suppression of cementite formation by Si [2]. Then, retained austenite often remains together with martensite and/or carbide at room temperature. This implies that austenite during isothermal heating is inhomogeneous and dynamically changing [3–5].

Controversy about the bainite transformation mechanism is still ongoing, and relates to whether the bainite transformation is essentially diffusional or displacive [6–12]. There are a number of studies utilizing in situ neutron or synchrotron X-ray diffraction experiments, and three-dimensional atom probe tomography (3D APT) to elucidate the mechanism of bainite transformation [2,3,6,13–16]. Their results indicate that neither the classic diffusional or displacive theory is sufficient, particularly to understand the behavior of carbon atoms in austenite.

The carbon enrichment in austenite increases the phase stability of austenite. Hence, the bainite transformation in Si-added steel sometimes halts before consuming all the austenite. This is called the incomplete transformation phenomenon. The two theories give different criteria for the interruption of bainite transformation. The displacive theory expects that the transformation halts when the Gibbs energies for bainitic ferrite and austenite are balanced with the same chemical composition. This is basically an idea in common with martensite transformation [11]. The diffusional theory, on the other hand, expects a local equilibrium to be reached around the phase boundary achieved by diffusion. The transformation halts when the driving force of the transformation balances with the energy dissipation during the phase boundary migration with the compositional contrast [17,18].

Therefore, the carbon enrichment process is the key to understanding the mechanism of bainite transformation. Specifically, the dynamic in situ observation for this mechanism should provide important knowledge about the behavior of carbon during the transformation.

The authors' group previously observed the anomalous shape of the diffraction peak for austenite during the isothermal heating of Fe-1.5Si-1.5Mn-0.15C steel by in situ neutron diffraction experiment [4]. We concluded that the peak was split into two peaks corresponding to the different carbon concentrations. The high carbon austenite, having ~1.2 mass% C, was retained even after cooling to room temperature, but the low carbon austenite transformed into martensite. This result indicated that the "to-be-retained" austenite was formed at an isothermal heating temperature and could be distinguished from the austenite that will transform to martensite.

Stone et al. [5] and Guo et al. [3] also observed similar peak splitting by in situ synchrotron X-ray experiments and concluded that a bimodal distribution carbon concentration existed. They explained that the high carbon austenite corresponds to the film-shaped retained austenite and low carbon austenite corresponds to the block-shaped martensite/austenite colonies. In contrast, in the above authors' study, a considerable amount of the retained austenite having block shape was detected. Hence, more detailed consideration of carbon diffusion during the isothermal heating should be attempted.

In this study, we conducted in situ neutron diffraction during the austempering process with various isothermal heating temperatures. Using a high carbon steel, Fe-1.9Si-1.0Mn-0.6C (mass%), a large amount of retained austenite was formed, enabling the detailed analyses. Together with the microstructural data, it was found that the nearly carbide-free microstructure having a large amount of retained austenite was formed only in a limited range of holding temperatures. Above and below the optimal temperature of 673 K, different mechanisms of carbide formation proceeded. The sensitive dependencies of the transformation behavior on the holding temperature indicated the importance of both the driving force of carbide precipitation and the rate of carbon diffusion.

## 2. Materials and Methods

### 2.1. In Situ Neutron Diffraction

In situ neutron diffraction experiments were conducted using the time-of-flight (TOF) neutron diffractometer, iMATERIA, which was built at the #20 beamline in the Materials and Life Science Facility of the Japan Proton Accelerator Research Complex (J-PARC MLF), Tokai, Japan. The infrared ray furnace equipped with helium gas injection nozzles was used

to perform the in situ measurement during heat treatment. The details of the instrument can be found in the authors' previous papers [4,19].

### 2.2. Analyses

iMATERIA is a versatile diffractometer equipped with numerous $^3$He position sensitive detectors covering a wide 2θ range [20]. The scattering data acquisition by all the detectors was conducted during the whole process of a heat treatment scheme. The scattering data acquired during the arbitral time span can be exported for the following analyses after the measurement. In the current study, we analyzed the data for every 200 s during the isothermal heating, in addition to the full austenizing step and after the final cooling to room temperature.

The lattice parameters of high and low carbon austenite were determined by fitting the 111γ peak by two overlapped Gaussian peaks on the diffractogram measured by the 90-degree detector bank ($76° < 2θ < 105°$). The carbon concentrations were calculated based on the obtained lattice parameters using the relationship suggested by Lee et al. [21]. After this, the phase fraction was determined by Rietveld texture analysis (RTA) using MAUD software (version 2.9.3, Luca Lutterotti, Trento, Italy).

The initial texture of the sample is indicated in Figure 1. The pole density is expressed as a multiple of that in a random distribution. The maximum pole density is 2.2 on the {111} pole figure, which is generally understood as a very weak texture. Note that the intensities of textures are presented by multiples of the random intensity in this study. However, this texture is not negligible for the quantitative phase fraction analysis as some of the authors previously proposed [22].

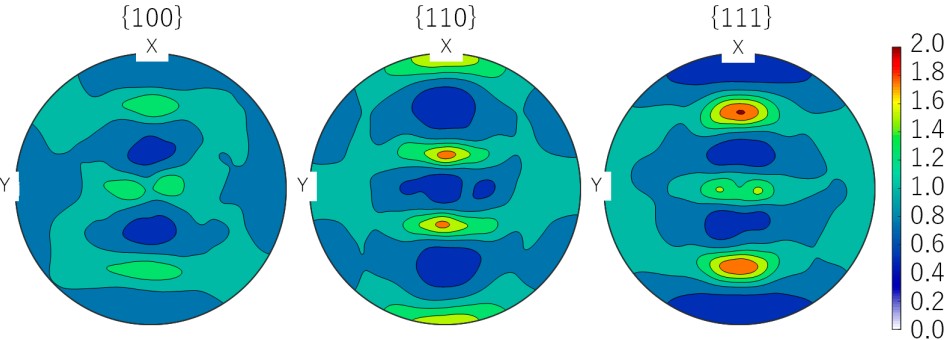

**Figure 1.** {100}, {110}, and {111} pole figures showing the initial texture of the tested steel. The directions X (vertical), Y (horizontal), and Z (vertical to the paper) correspond to RD (rolling direction), TD (transverse direction), and ND (normal direction) in the hot-rolling process, respectively.

Since RTA can introduce the appropriate correction for preferential orientation by considering perfect information of textures, quantitative evaluation was realized. To avoid the instability of the Rietveld analysis, the lattice parameters for the low and high carbon austenite were fixed with the values obtained by the above Gaussian fitting analysis. The details of RTA can be found elsewhere [22–24].

### 2.3. Sample

The sample used in this study was Fe-1.9Si-1.0Mn-0.6C (mass%). The detailed composition is listed in Table 1. The vacuum-cast ingot was hot forged after reheating to 1473 K and hot rolled, resulting in a final thickness of 4 mm. The preheating prior to the hot rolling was conducted at 1423 K for 3.6 ks. The sample was quenched after the rolling by high-pressure water spray to 873 K to prevent the coarsening of the microstructure. Then, the sample was gradually cooled to room temperature in a furnace. Both sides of the surface regions were mechanically removed and a sample sheet having a thickness of 2 mm was prepared. The sample shape was the same as in the previous research, roughly having a shape of 70 mm in the rolling direction (RD) × 10 mm in the transverse direction

(TD) × 2 mm of sheet thickness [4]. The microstructure consisted of tempered pearlite and primary ferrite (dark regions), as shown in Figure 2. t

**Table 1.** Chemical composition of the sample in mass%.

| C | Si | Mn | P | S | Al | N |
|---|---|---|---|---|---|---|
| 0.61 | 1.90 | 0.98 | 0.008 | 0.001 | 0.033 | 0.04 |

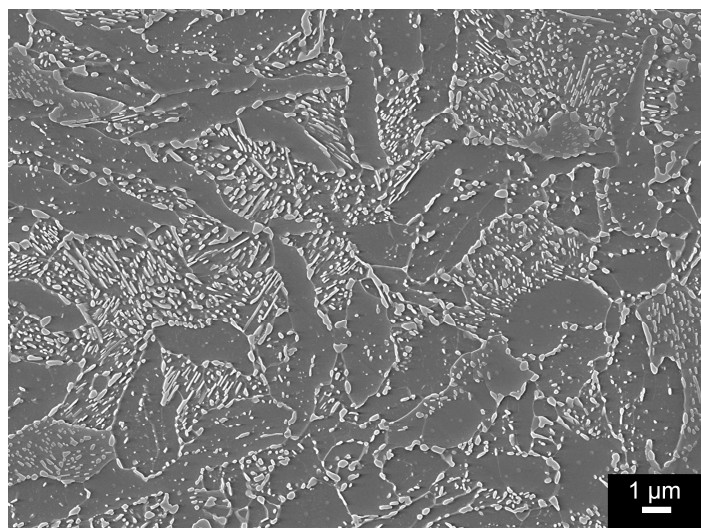

**Figure 2.** Initial microstructure of the current sample (secondary electron image, acceleration voltage: 15.0 kV, working distance: 15.0 mm). The horizontal and vertical directions are the RD and ND in the hot-rolling process, respectively.

*2.4. Heat Treatment Scheme*

The heating schemes applied in this study are shown in Figure 3. The curves are the recorded temperature histories during the in situ neutron diffraction experiments. Firstly, the sample was heated to 973 K and kept for 300 s until there was no change in phase composition just before the transformation to austenite. The sample was then fully austenitized at 1173 K and gas-quenched at the rate of about 20 K/s to the isothermal holding temperature and kept for 1800 s at 573, 623, 673, 773, and 923 K, and for 3600 s at 723 K. The longer holding time only at 723 K was due to the slow reaction observed in this condition. An additional experiment was also conducted with holding for 100 s at 673 K to observe the transient microstructures.

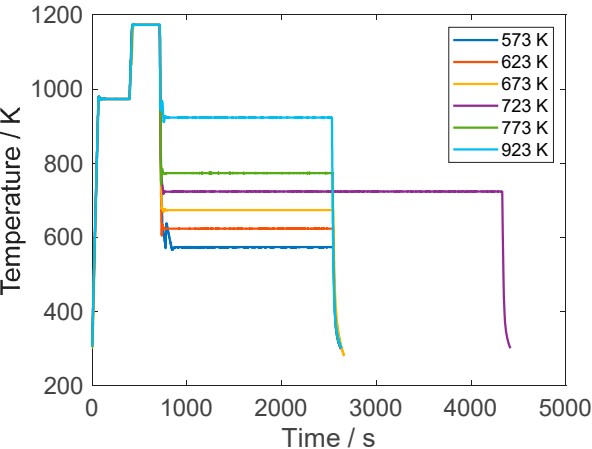

**Figure 3.** Heating schemes applied in this study.

### 2.5. Microstructural Observation

The microstructure was observed by both secondary electron imaging and electron back scattering diffraction (EBSD) using a field emission scanning electron microscope. The former was beneficial to find carbides and latter aided observation of the shapes of bainite units and retained austenite. The step size for the EBSD measurement was 50 nm. The observation was conducted on the mid-thickness area on a cross-section perpendicular to the transverse direction in the hot-rolling process.

## 3. Results

### 3.1. Dynamic Change in Phase Fractions and Carbon Distribution

As described in the introduction, the irregular shape of the diffraction peak for austenite during the isothermal heating is of central interest. As shown in Figure 4, the 111 diffraction peak for austenite observed in this study can be understood as the overlap of two Gaussian peaks. Since the lattice parameter of the austenite is sensitive to the carbon concentration, this result indicates that the carbon concentration distribution was not uniform but had a bimodal distribution. To take into account the peak splitting in Rietveld texture analysis, we consider that there were two different "phases" having the FCC crystal structure but different lattice parameters. That having the larger lattice parameter is called high carbon austenite and the other is low carbon austenite. Hence, the phase fraction analysis was carried out as the competition among three phases: two austenite and BCC ferrite.

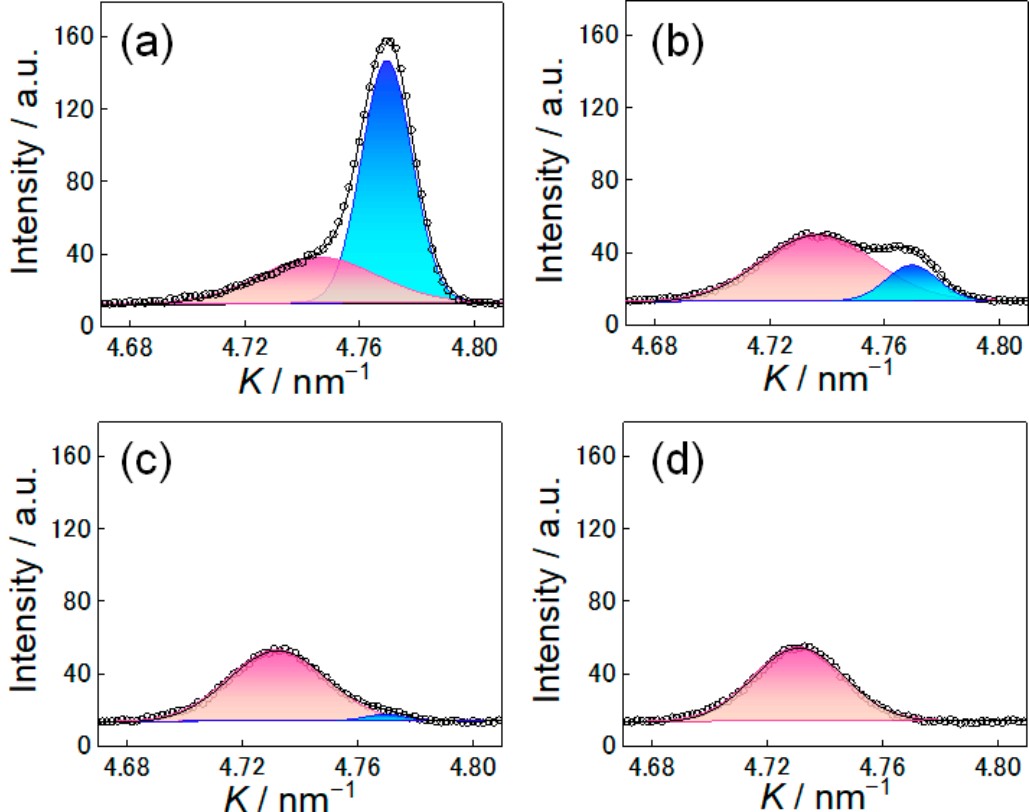

**Figure 4.** The splitting of the $111\gamma$ diffraction peak observed during the isothermal heating at 673 K. The images correspond to different time spans, (**a**) 0−200 s, (**b**) 200−400 s, (**c**) 400−600 s, and (**d**) 600−800 s. The origin of the time is the beginning of isothermal heating. The observed intensities are shown as open circles tied with black lines. The split peak observed at each time span was fitted by pink and blue Gaussian peaks representing the high and low carbon austenite, respectively. The magnitude of the scattering vector, *K*, is defined as $K = 1/d = 2\sin\theta/\lambda$.

In this series of diffraction experiments, the peaks for cementite or any carbide were not visible during and after isothermal holding at 773 K or lower. However, the micrographs indicated certain amounts of carbide should be formed. The reason for this confliction is discussed in the following section.

Figure 5 shows the changes in carbon contents in high and low carbon austenite phases and phase fractions during isothermal heating at different temperatures analyzed during the in situ neutron experiments. The result at 973 K is not because pearlite transformation occurred immediately after reaching to the holding temperature. The analytical fitting errors are as small as the points for carbon concentrations. The analytical errors for phase fractions are also very small, i.e., usually less than 1%. However, this is true only if all the assumption/modeling taken into account for the RTA are appropriate. Based on the previous verification for the precision of phase fraction analysis [22], the error relative to the true phase fraction should be less than 5%.

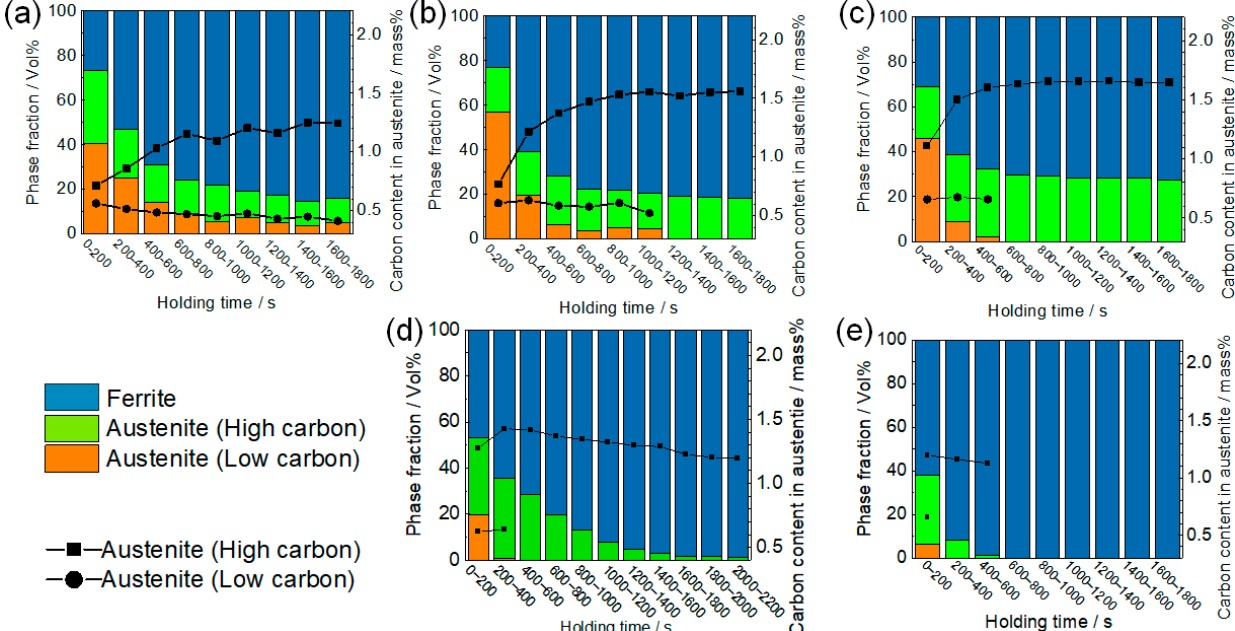

**Figure 5.** The phase fractions and carbon concentrations in high and low carbon austenite during austempering at various temperatures: (**a**) 573 K, (**b**) 623 K, (**c**) 673 K, (**d**) 723 K, and (**e**) 773 K.

Among the temperatures, it is found that the low carbon austenite has a carbon concentration close to 0.6 mass%, which is the bulk composition. This means that the expelled carbon from bainite did not spread rapidly but formed the high concentration regions. The carbon concentration in the high carbon austenite gradually increased up to around 1.6 mass% at 623 and 673 K. In these conditions, the high carbon austenite is quite stable. Even after the low carbon austenite is fully consumed, the high carbon austenite remained, i.e., the so-called incomplete reaction phenomenon [12].

One may note that the lattice parameter of austenite can be affected not only by carbon concentration, but also the elastic strain caused by the transformation. In the current evaluation, the increase of 0.1 mass% carbon corresponds to $8.6 \times 10^{-4}$ of lattice strain. By roughly assuming a Young's modulus of 200 GPa, and that lattice strain is due to the internal stress, the magnitude of internal stress would be 170 MPa. Since the yield stress of 0.6 C austenitic steel seems to be around 200 MPa [25], the internal stress higher than this can plastically be accommodated. Therefore, even if an internal elastic stress field existed, the error of carbon concentration would be only as high as 0.1 mass%.

The time required to achieve the saturation of carbon concentration became shorter at higher temperatures. However, above 723 K, the carbon concentration decreased with increasing time. The phase fraction of high carbon austenite also decreased. Contrarily, the

fraction and carbon content in high carbon austenite continued to increase or stabilized below 673 K.

The low carbon austenite can survive longer at lower holding temperatures although the driving force of the $\gamma \rightarrow \alpha$ transformation should be larger. This indicates that the rate of bainite transformation is not controlled by the driving force, but is more likely influenced by diffusion-related processes.

The fraction of high carbon austenite, or retained austenite after quenching, was highest at 673 K. When the temperature was too high or too low compared to this optimal value, a lower amount of retained austenite was found after the heat treatment.

### 3.2. Microstructural Features

Figure 6 shows the image quality (IQ) maps based on the EBSD measurements for the samples after the austempering experiments. Austenite is colored red and the bright gray regions represent ferrite. The dark IQ regions are boundaries, cementite, or martensite.

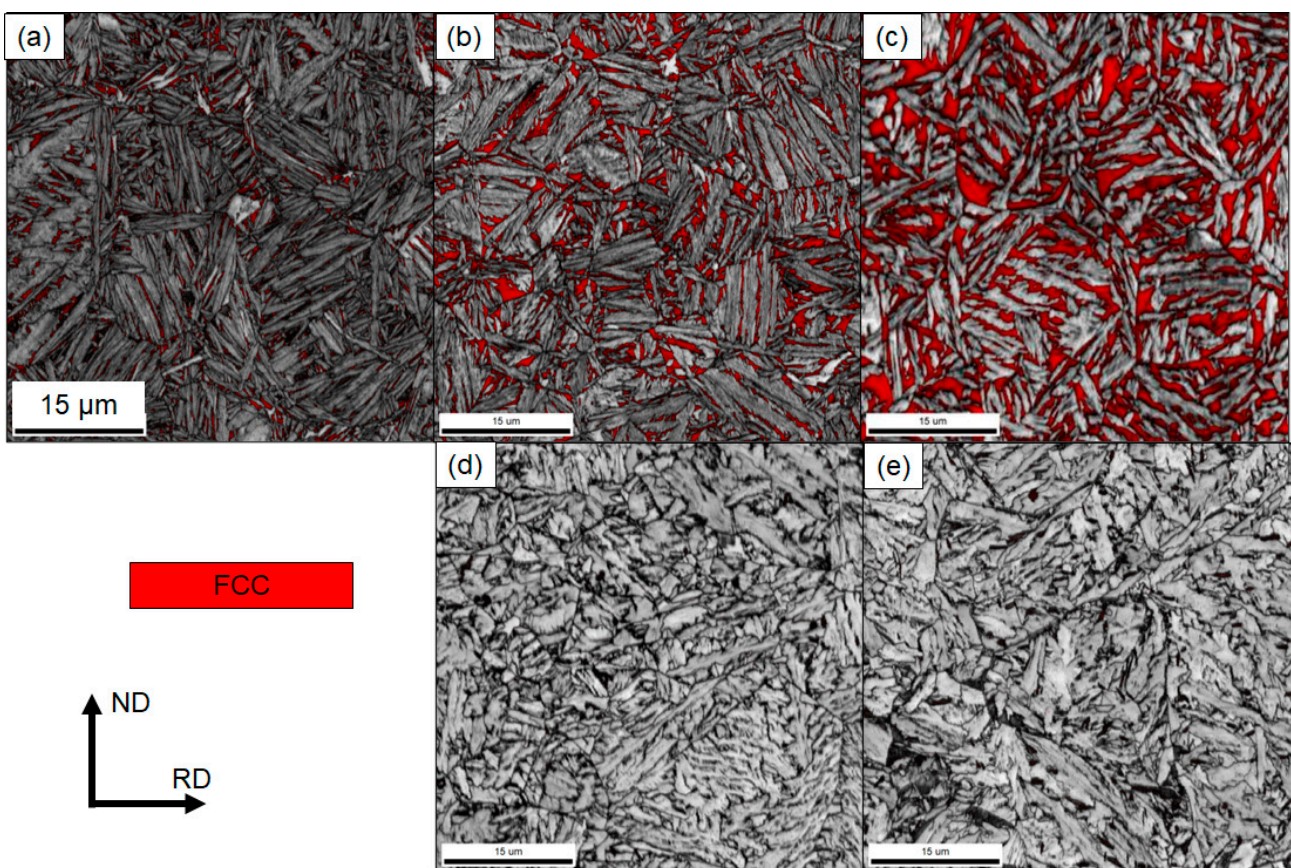

**Figure 6.** Image quality (IQ) map with overlayed phase coloring, where red regions are FCC austenite. The samples are after the heat treatment at (**a**) 573 K for 1.8 ks, (**b**) 623 K for 1.8 ks, (**c**) 673 K for 1.8 ks, (**d**) 723 K for 3.6 ks, and (**e**) 773 K for 1.8 ks.

A considerable amount of austenite was detected for the samples treated at 673 K or lower (Figure 6a–c). At lower temperatures, in Figure 6a,b, the bainitic ferrite sheaves are confirmed as flat and parallel bands. Retained austenite is seen these bands. In Figure 6c, coarser ferrite laths are more evident than in (a) and (b), resulting in the serrated phase interfaces. However, the sandwich structure consisting of parallel ferrite sheaves and retained austenite is the common feature among these three microstructures.

As expected from the neutron diffraction results, austenite is merely detected in Figure 6d,e. The dark regions correspond to cementite or fine pearlite. These consist of mostly lath-shaped ferrite, but parallel sheaf bands are rarely seen.

All of the microstructures also have submicron-scale features, as shown in Figure 7, which may not be captured by EBSD due to its nature of stepping measurement with a finite probe size. In these secondary electron images, the dark, recessed regions are ferrite, and the remaining white-rimmed zones are retained austenite, cementite, or martensite. Although it is difficult to distinguish them accurately, the morphological features contain some information.

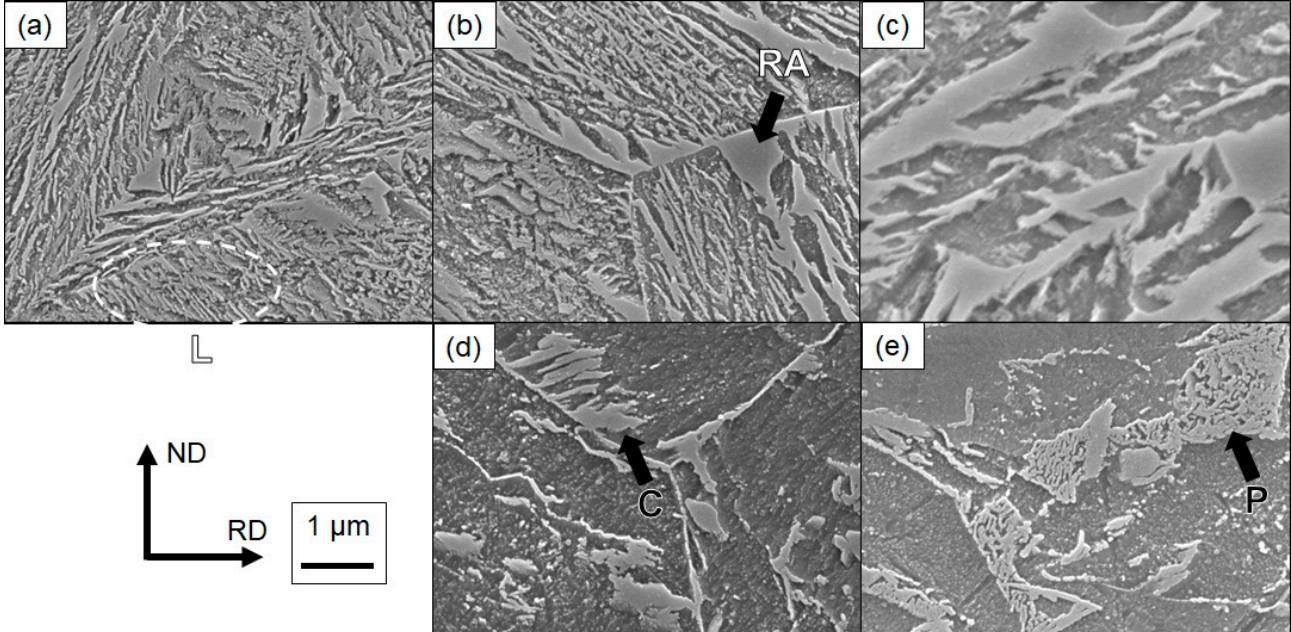

**Figure 7.** Secondary electron images (20k × magnification) for the samples after the heat treatment at (**a**) 573 K for 1.8 ks, (**b**) 623 K for 1.8 ks, (**c**) 673 K for 1.8 ks, (**d**) 723 K for 3.6 ks, and (**e**) 773 K for 1.8 ks.

In Figure 7a–c, the white blocky features should mainly be retained (high carbon) austenite (RA in Figure 7b, for example). Some martensite, which corresponds to the low carbon austenite during the isothermal heating, are included in Figure 7a. Together, the parallel alignment of thin film-like particles can be confirmed, as shown in the enclosed area, L. This microstructure is similar to the lamellar carbide in lower bainite seen in steels with no silicon or low amount [10,26]. The similar film-like particles can also be seen in Figure 7b,c. Further discussion of this feature is presented in the next section.

In Figure 7d,e, most of the particles, such as that indicated by C, should be cementite. In addition to the blocky cementite, the thin films decorating prior austenite grain boundaries and bainite sheaf boundaries are observed. In Figure 7e, the fine pearlite-like structure can also be confirmed, as shown as P.

## 4. Discussion

### 4.1. Carbide/Austenite Nano-Films

Although considerable amounts of carbide were confirmed in microstructures, as shown in Figure 7, as mentioned above, carbide was not detected in neutron diffraction. An explanation should be provided for this conflict.

First, we discuss the results at higher temperatures, 723 and 773 K, where no austenite was confirmed after the heat treatments. Figure 8 shows the diffraction patterns for the same sample but measured before and after the isothermal heating heat treatment at 773 K. The cementite peaks were obvious before the treatment (see Figure 2 for the microstructure), but they disappeared after the isothermal heating. Because there is no austenite, most of the carbon should exist as carbide.

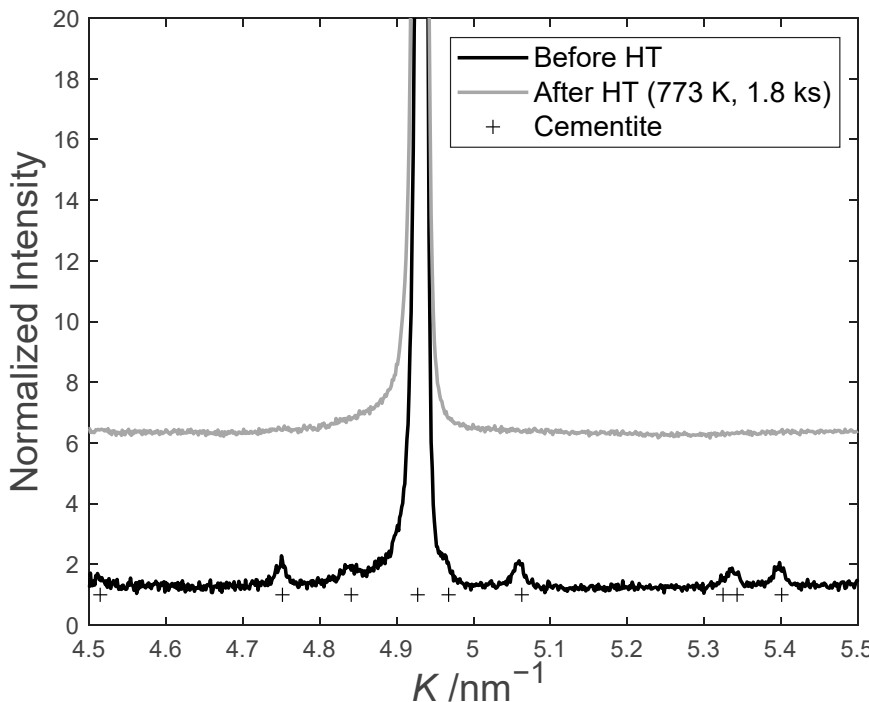

**Figure 8.** Diffraction patterns near the 110α peak observed before (black) and after (gray) the heat treatment process with austempering at 773 K for 1.8 ks. The cementite peak positions are indicated by "+".

We understand that the lack of diffraction peaks of carbide is due to the small coherent crystallite size resulting from the very limited growth rate in transformation. If the crystallite size was several tens of nanometers, severe peak broadening would suppress the peak height. If the broadening effect is severe, the peak height can be lower than the limit of detection. In the study of the plastic deformation of pearlitic steel, the severe peak broadening is often observed, even at small strain values [27]. In addition, the current instrumental arrangement may also promote the peak broadening. The Scherrer equation, $\beta = C\lambda / D_c \cos \theta$, is the fundamental relationship between the crystallite size, $D_c$, and peak broadening, $\beta$ [28,29]. C is a constant, $\lambda$ is the wavelength, and $\theta$ is the scattering angle. With the current instrumental condition ($\lambda \cong 0.38$ nm and $\theta \cong 77°$ for the backscattering bank and $K$ range shown in Figure 8), $\lambda/\cos \theta$ is around ten times as large as ordinary X-ray diffraction ($\lambda = 0.154$ nm and $\theta \cong 45°$). The transmission electron diffraction experiment applies a much shorter wavelength and smaller diffraction angle. Thus, more severe peak broadening is expected for the current setting than in transmission X-ray or electron diffraction techniques. It should be admitted that the current technique is not capable of the detection of fine carbide. Although iMATERIA is also equipped with low $2\theta$ angle detectors, the resolution of $K$ worsens with decreasing $\theta$ [20]. The neutrons corresponding to the $K$ range of interest are too fast to resolve $\lambda$ by time-of-flight measurement.

In turn, it is strongly inferred that the bainitic cementite may consist of fine crystallites, although the apparent size of them in the micrographs (Figure 7) is not much different from that in pearlite (Figure 2).

It is therefore noted that the absence of diffraction peaks does not necessarily mean the absence of the phase. Therefore, the microstructure of the bainitic steels must be studied not only by neutron diffraction, but also by microscopic techniques.

Second, we discuss the feathery structure consisting of nano-thickness plates seen at lower temperatures (Figure 7a–c). The plate shape particles can be either austenite or cementite. Chang reported a similar microstructure having thin austenite between the ferrite subunits by transmission electron microscopy for 2.10Si-2.15Mn-0.46C steel [30]. Suzuki et al. also showed a scanning electron micrograph that was similar to that of the current study [26].

They studied JIS-SUP7 steel, whose composition (2.00Si-0.75Mn-0.57C + impurities) was close to that of the current sample, and heat treatment conditions were also similar to ours. For both austenite and cementite, they detected the thin particles formed by the isothermal heating at 573 K by selected area diffraction using a transmission electron microscope. They concluded that it was difficult to distinguish whether a thin particle was cementite or austenite from morphological aspect.

### 4.2. Carbon Migration Mechanisms

To evaluate the fraction of carbon stored in such "invisible" fine carbide/austenite film, we determined the fraction of carbon in detected austenite (FCDA) as follows:

$$\text{FCDA} = \left( X_{\gamma-\text{High C}} V_{\gamma-\text{High C}} + X_{\gamma-\text{low C}} V_{\gamma-\text{low C}} \right) / X_{\text{bulk}} \tag{1}$$

where $X_{\gamma-\text{High C}}$ is the mass carbon concentration in high carbon austenite, and $V_{\gamma-\text{High C}}$ is the volume fraction (regarded as mostly the same as mass fraction) of high carbon austenite. The same definitions are applied for low carbon austenite ($\gamma$-Low C). $X_{\text{bulk}}$ is the bulk concentration of carbon, 0.61 mass%. These values were measured by the neutron diffraction and shown in Figure 4. If FCDA = 1, all the carbon atoms dissolve into block-shape austenite. However, if FCDA < 1, the remainder (1 − FCDA) should exist as film-shape particles or in ferrite. However, the detailed analysis for the ferrite peaks did not show any evidence of supersaturation of carbon, i.e., change in lattice parameter or asymmetry resulting from the lattice tetragonality. Hence, we believe that most of the undetected carbon exists as carbide/austenite nano-films.

The changes in FCDA during the isothermal heating at three different temperatures are shown in Figure 9. At 573 K, FCDA decreases with time. It is understood that the expelled carbon from the transformed region is mostly trapped in the vicinity of the ferrite sub-unit, forming the film austenite or cementite. As explained above, these cannot be detected by the current diffraction experiment. Such film formation should be due to the low diffusion rate of carbon in austenite. Therefore, the formation of bulky high carbon austenite, and the increase in its carbon concentration, are slow.

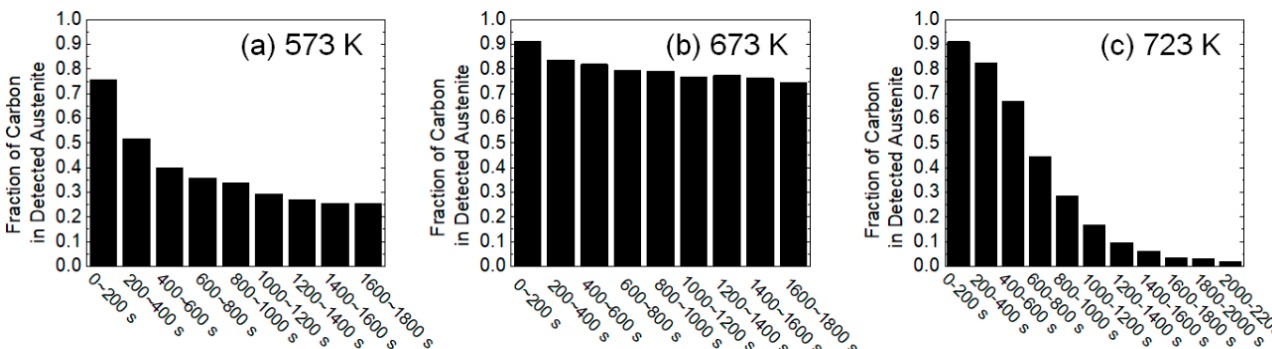

**Figure 9.** The fraction of carbon in detected austenite (FCDA) during the isothermal heating at (**a**) 573 K, (**b**) 673 K, and (**c**) 723 K.

In contrast, at 673 K, FCDA remains higher than 0.7, although the notable decrease in low carbon austenite below 600 s is confirmed in Figure 5c. This means that the carbon atoms migrate from disappearing austenite to the remaining austenite region, resulting in the formation of high carbon block-shape austenite, which is stable during the isothermal heating.

The similar carbon enrichment occurs at the first stage of annealing at 723 K, resulting in the formation of high carbon austenite. The FCDA remains high during the first 400 s but decreases with decreasing fraction of high carbon austenite. This is due to the cementite precipitation from carbon-enriched austenite. The carbon-depleted zone around the

cementite can be the nucleation site of bainitic ferrite. Therefore, the final microstructure only contains bainitic ferrite and cementite.

The bainite microstructure is often categorized into two groups, i.e., lower and upper bainite. Although some different definitions for the categorization are suggested, we mainly follow the definitions based on the carbide morphology suggested by Takahashi and Bhadeshia [31]. In lower bainite, according to Takahashi et al., fine carbide platelets are seen inside of the bainite "plate". It is unclear if the "plate" means sub-unit or sheaf, but they considered that the carbide was precipitated from supersaturated ferrite. The feathery carbide distributions seen in Figure 7a,b correspond to this feature. However, at the same time, these microstructure looks like lower bainite with lamellar carbides, as noted by Borgenstam et al. [10]. Since the bimodal carbon distribution and high FCDA was achieved during a very early stage of the isothermal heating in the current result, the carbon migration to austenite should take place immediately after that. It is hence difficult to imagine that carbon remained in ferrite and formed carbide in it. The sequential formation of a fine ferrite sub-unit and carbide suggested by Ohmori et al. [14] sounds like a more reasonable explanation. By following this explanation, however, the microstructure is categorized as "B-III upper bainite".

Furthermore, in the upper bainite, as defined by Takahashi and Bhadeshia, the sheaves merely include carbide, but film cementite exists between the sheaves, in the case of low carbon plain steels. The carbide films (seen as white lines) can be confirmed in Figure 7d,e. The carbon-enriched zone can be retained as austenite in the case of Si-added steels, as seen in this study. Figure 6b shows the existence of film austenite between the sheaves. Therefore, the microstructure formed at 623 K has the features of both lower and upper bainite.

The exceptional microstructure shown in Figure 6c has a large amount of austenite. The shape of retained austenite is no longer film in the TEM order, but has a thickness of around 0.3 µm. Correspondingly, as shown in Figure 9b, a large fraction of carbon is enriched in austenite.

### 4.3. Formation of High Carbon Austenite at 673 K

Figure 10 shows the microstructure obtained by quenching after the isothermal heating at 673 K for 100 s. The matrix is martensite, which corresponds to the low carbon austenite before cooling to room temperature. However, the bainitic ferrite sheaves indicated by the arrows, and retained austenite, are easily distinguished from the martensitic matrix. This image agrees with the schematic drawings of upper bainite shown by Timokhina et al. [16].

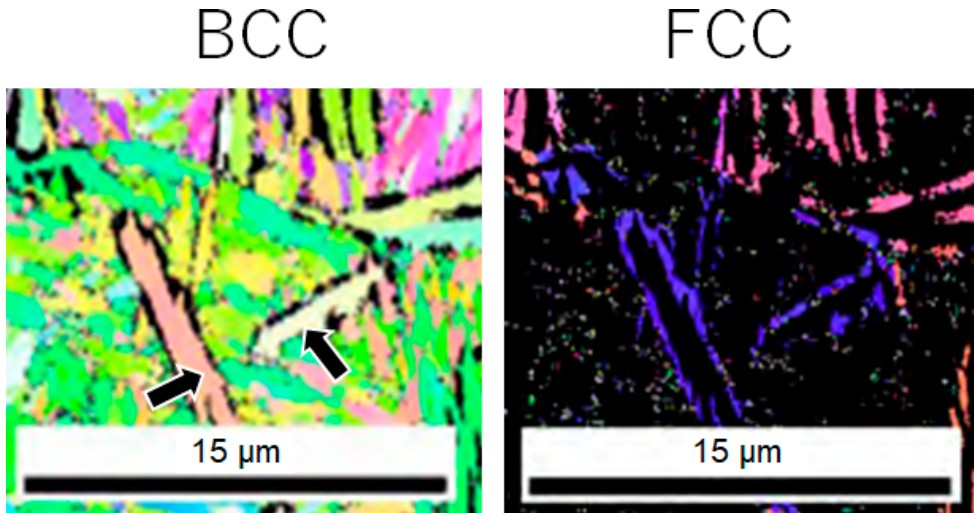

**Figure 10.** EBSD orientation maps for BCC (ferrite/martensite) and FCC (retained austenite) for the sample after isothermal heating at 673 K for 100 s. The arrows indicate bainitic ferrite sheaves.

As clearly seen here, the isolated sheaf is surrounded by the retained austenite layer. Guo et al. explained that the high carbon film austenite is formed as the result of enclosure by ferrite sheaves or sub-units [3]. Since the trapped carbon between the ferrite sub-units cannot escape to the matrix, i.e., low carbon austenite, the bimodal distribution of carbon is achieved. They explained that this was also the reason the carbon concentration in the high carbon austenite exceeds the value expected by the $T_o$ line. However, the current situation conflicts with this explanation; the high carbon austenite layer exists adjacent to the matrix. Timokhina et al. suggested that dislocations can trap carbon, based on their TEM and 3D atom probe tomography (3D APT) experiments. In Figure 10, the thickness of the ferrite sheaf is around 1 μm, which is almost the same as that in Figure 6c. Therefore, the following transformation cycle can be suggested: (i) the sheaf, cluster of ferrite sub-units is formed in relatively short time; (ii) the stress field caused by transformation strain develops around the sheaf, which is accommodated by dislocations; and (iii) the carbon is trapped by the dislocations around the sheaf, resulting in the formation of the high carbon austenite region. The transformation proceeds by increasing the number of the ferrite sheaves surrounded by the high carbon regions. The geometrical enrichment suggested by Guo et al. may further increase the carbon concentration in the high carbon austenite if the parallel sheaves are developed and austenite is sandwiched by them. However, this is not the fundamental reason for the formation of high carbon austenite.

This mechanism may be applied regardless of temperature, but another important factor is the precipitation of carbide. Above 723 K, the diffusion of Si enables the carbide precipitation, so the high carbon austenite is no longer stable. The pearlite structure seen in Figure 7e represents good evidence that the diffusion of Fe and substitutional elements is available at this temperature. The austenite grain boundary can be the nucleation site of both ferrite and cementite in this case, resulting in the decoration of the grain boundary with cementite walls seen in Figure 7d,e.

At lower temperatures, 573 and 623 K, the rate of diffusion of Si is not sufficient but Si containing cementite becomes stable in this range. In Figure 11, the maximum carbon concentrations in austenite observed in this study are plotted by black squares, together with para Ae3, para $A_{cm}$, $T_o$, and WBs curves. Para $Ae_3$, para $A_{cm}$, and $T_o$ curves were calculated by Thermo-Calc software and the WBs line is based on the experimental fitting by Leach et al. [32]. Both $T_o$ and WBs are basically lines from left-top to right-bottom, but the para $A_{cm}$ curve has a positive tangent. The carbon concentrations seen at 573 and 623 K are quite close to the para $A_{cm}$ curve. Therefore, it is concluded that there is a chemical driving force for the cementite precipitation in this temperature range, even though diffusion of Si is prohibited.

Regarding the higher temperatures, the maximum carbon concentrations in austenite are higher than those expected by $T_o$ but lower than the para $A_{cm}$ concentration. Thus, incomplete bainite transformation took place but the stasis cannot be explained by the displacive theory. Furthermore, nor does the WBs line fit well, although the tangent is somewhat close. In Leach's study, the coefficients for Mn and Si were determined with very scattered data compared to those for C. Since WBs is the phenomenological quantity, it is difficult to identify what is appropriate or not.

A possible explanation for this is that the energy dissipation during the boundary migration limits the rate of transformation [17]. The introduction of the energy dissipation as the additional driving force of transformation results in the shift of para Ae3 toward the left in Figure 11. Several sources of energy dissipation are suggested, i.e., solute dragging, intrinsic mobility, element partitioning, and transformation strain. Wu et al. suggested that the contribution of transformation strain energy can be the dominant source of energy dissipation in the case of bainite transformation of Si-added steel [33]. Moreover, it should be noted that the carbon concentrations observed in this study are the averages of certain volumes. It is possible that the local equilibrium is established with a higher carbon concentration in the vicinity of phase boundaries.

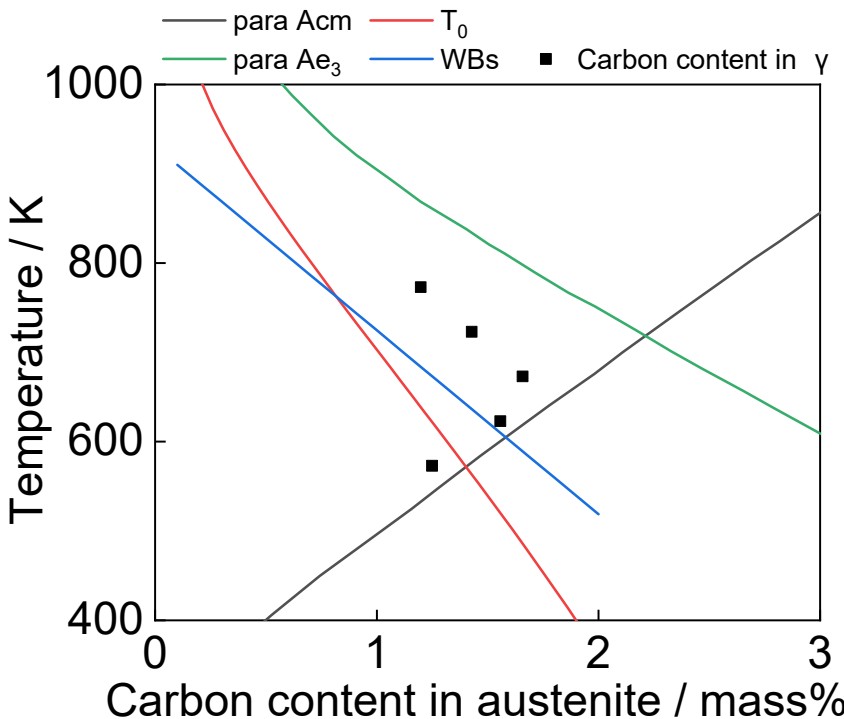

**Figure 11.** Para equilibrium phase diagram calculated by Thermo-Calc. The WBs line is drawn using the equation suggested by Leach et al. [31]. The maximum carbon concentrations in austenite determined in the current experiment are plotted together.

## 5. Conclusions

In situ neutron diffraction experiments were conducted for Fe-0.61C-1.9Si-0.98Mn during the austempering processes, i.e., isothermal heating at 573–773 K after full austenizing. Bainite transformation behavior varied depending on the holding temperature. Correspondingly, different behaviors of carbon enrichment in austenite were observed. The carbon concentration in austenite during isothermal heating showed a bimodal distribution at every tested temperature. The high carbon austenite had a carbon concentration that was higher than 1 mass%, whereas the remaining low carbon regions had around 0.6 mass%, which corresponded to the bulk concentration. The bainite transformation proceeded by consuming the low carbon austenite.

The highest amount of retained austenite was formed by holding the temperature at 673 K. The high carbon austenite surrounded the bainitic ferrite sheaves, but was not necessarily between the sheaves. The cause of the formation of the carbon concentration gradient between high and low carbon austenite regions was not clear, but it may be possible that the local strain field and/or dislocations captured carbon atoms in the vicinity of bainitic ferrite. After the depletion of low carbon austenite, neither the phase fraction or carbon concentration in austenite changed during the isothermal heating for 1800 s.

At the temperatures of 723 and 773 K, cementite precipitation from high carbon austenite was observed. This should be due to the considerable rate of diffusion of Si. Therefore, austenite was completely transformed into ferrite and cementite. By comparison, at temperatures of 573 and 623 K, carbide precipitation was possible without the diffusion of Fe and the substitutional elements as expected by the para equilibrium phase diagram. The maximum carbon concentration in high carbon austenite agreed well with the para $A_{cm}$ curve in the phase diagram.

Above 673 K, however, the maximum carbon densities in high carbon austenite were considerably higher than the expectations formed from the $T_0$ curve. This suggests that the transformation proceeded with a diffusional nature. However, the energy dissipation

during the phase boundary migration should be considered to explain the deviation of the observed carbon concentration from the para Ae$_3$ curve.

**Author Contributions:** Conceptualization, K.F., Y.T., T.T. and K.K.; Formal analysis, Y.O. and K.U.; Investigation, Y.O., K.U., K.F. and Y.T.; Methodology, Y.O., K.U. and S.S.; Supervision, T.T., K.K. and S.S.; Validation, Y.O., T.T. and S.S.; Visualization, K.U.; Writing—original draft, Y.O.; Writing—review and editing, Y.O., K.F., Y.T., T.T., K.K. and S.S. All authors have read and agreed to the published version of the manuscript.

**Funding:** This research received no external funding.

**Institutional Review Board Statement:** Not applicable.

**Informed Consent Statement:** Not applicable.

**Data Availability Statement:** Not applicable.

**Acknowledgments:** The neutron experiments at the Materials and Life Science Experimental Facility (MLF) of the J-PARC were performed under user programs (Proposal No. 2017BM0023, 2018BM0029, 2019PM3002, and 2020PM3001).

**Conflicts of Interest:** The authors declare no conflict of interest.

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
