# Peer review of "Microstructure Formation and Carbon Partitioning with Austenite Decomposition during Isothermal Heating Process in Fe-Si-Mn-C Steel Monitored by In Situ Time-of-Flight Neutron Diffraction"

_metals, doi:10.3390/met12060957_

Round 1
Reviewer 1 Report
The presented text describes the results of the study of the microstructure formation in the austempered steel at various temperatures using mainly neutron diffraction and electron microscopy.
The topic is very interesting for industry users. The experiments are well prepared and conducted. The different analyses are well described. The results are presented clearly and in order. Below are some comments and thoughts about the conclusions presented within the text.
On lines 122-125, the authors claim the texture density, what is the unit of it? The previously reported works about wrong phase fractions should have some references.
In Figure 1 should be shown what are the main directions as ND, TD, and RD in correspondence to x, y and z.
In Figure 4, what is the x-axis value K? The d-spacing of austenite 111 reflection is around 0.209 nm, scattering vector Q corresponds to 30.01 nm-1. From the figure is evident that both reflections have different broadening most probably caused by the various microstrain of the austenities. Did you try to evaluate as well the microstructure based on the peak broadening? Inappropriate microstructure handling can also cause the wrong phase fraction evaluation.
On line 197, there is written that “the carbon density close to 0.6 mass%”. Do you mean the carbon concentration or fraction rather than the density? This is mentioned several times in the further text.
It is not clear which part of the structure is shown in Figure 7. It is related to the dark or light sub-structures?
The description in chapter 3.2 is very confusing. The authors should be able to distinguish between cementite, ferrite or austenite (line 305, 336). Trying to conclude the phase content only by morphology can be quite misleading.
In lines 272-276, are described the results from the “higher temperatures”, but mentioned are the lower ones. And even though there are no data for 523 K. The temperature of 523 K appears several times in the further text.
It would be good to present the error bars in Figure 5. How precise is the evaluation of the carbon content from the cell parameters and the phase fraction?
The text in chapter 4.2 is difficult to follow. Unclear explanation of underlining process. For example that carbon expelled from the transformed region forms the austenite (line 336). It is not clear the outcome of the discussion.
Do you really think that the carbon is trapped by dislocations (line 394)? Usually, the dislocations are trapped by carbon.
In conclusion, the text will need some deep rework to consolidate very interesting findings and provide a more clear description of the achieved understanding.
Author Response
Thank you for your review. Please find my answers for your comments in the attached document.

Reviewer 2 Report
Comments to the manuscript by Y. Onuki et al. Metals 1678214.
This manuscript presents an experimental study on monitoring evolutions of fraction of phases, lattice parameters and carbon concentrations during isothermal transformation of austenite decomposition using in-situ neutron diffraction. The experimental data may enrich researches in the area of bainitic transformation, whose essential mechanism remains unresolved. However, interpretation of the experimental data is not sufficiently clear, sometimes contradictory and not properly explained. Partly this is due to the writing (text has to be polished), while some physics have also been inappropriately understood. Therefore, I recommend to have this manuscript substantially revised by the authors before it can be considered further. Specific comments are as follows.
1) Some terminologies are not properly used. Examples: “dynamic phase fraction”; what does it mean “chemical stability” specifically? “Austempering”, I would rather recommend to describe it as austenite decomposition during isothermal heat treatment or something similar; ”carbon density”, should not be “carbon concentration” better?
2) Line 54-55, this is a very biased statement. So far, both the diffusional and diffusionless views are respectively supported by different experimental observations, whereas they both cannot fully explain all the observations. It is of little use/interest to claim which is more obvious. In addition, in ref. 16, as it was mentioned in the Abstract, “formation of bainitic ferrite nuclei with carbon-depleted regions…” supports the diffusional theory.
3) The introduction should be more concise. Recent advances about the effect of solutes on migration of austenite-ferrite/bainite interfaces should be commented and referenced.
4) Can you explain in more detail about how the initial “weak” texture was considered into the Rietveld fittings?
5) Section 2.4. How the sample temperature was measured? Figure 3 shows that there is a temperature oscillation in the early stage of isothermal heat treatment. This is fine, however, instead of having an undershoot, there is an overshoot, which is confusing. Can you explain and probably show zoom-in plots for the temperature profiles at this stage.
6) How the lattice parameters of bainitic ferrite were determined? Any assumption for its carbon concentration?
7) Line 272-277, the text is confusing. 523 and 573 K are higher or lower temperatures here? “no austenite and carbon should exist as carbide”? Did you mean there is no retained austenite as shown in Figure 6d and 6e, thus the carbon is expected to be in cementite? However, no diffraction peaks of cementite are observed experimentally. The authors should be able to derive the volume fractions of cementite based on the mass balance of carbon for the higher temperature cases. If it is not significantly low, we should expect to see diffraction peaks of cementite. Otherwise, there may be wrong assumptions in the data analysis.
8) Line 283-290, the text is also confusing and it is not helpful to explain the peak broadening.
9) Equation 1 suggests an assumption that carbon concentration in bainite is zero and there is no density difference between bainite and austenite.
10) Line 331, what does “particles” specifically mean?
11) Figure 11. The current interpretation is perhaps not physically sound. Here is another way of interpretation. The data points for higher temperatures (773 and 723) are located between the para Ae3 line and the T0 or WBs lines, while the data points for lower temperatures (573 and 623 K) are located along the para-Acm line, and the data for 673 K is a transition point. Staying a bit away from the para-equilibrium lines is strong evidence for “incomplete transformation” or “transformation stasis”, which is a very common phonomenon in bainitic transformation. Now, the C concentrations overpass the equilibrium values predicted by T0 model. This suggests a diffusional nature and according to the diffusional theory, the “incomplete transformation” is mainly caused by the solute segregation at the interface, i.e. solute drag. Some recent articles have used the solute drag based models, e.g. the Gibbs energy balance model, to explain this phenomenon. Both the experimental and modelling results show that the predicted bainitic fraction is located in between the T0 line and the para-Ae3 line, similar to what is shown in Figure 11 for the maximum carbon concentration in the austenite.
Another point needs to be noted. Since neutron diffraction cannot resolve carbon concentration spatially, the experimental data points plotted in Figure 11 underestimated the actual maximum carbon concentration in the austenite, i.e. the maximum concentration of the diffusion profile if there exists. Because it has to average the C concentrations over the entire high-carbon austenite regions based on fittings for the diffraction peaks.
As for the lower temperature data, carbon diffusion is thermodynamically limited by the driving force of the cementite.
I suggest the authors to rethink and probably to rewrite this part.
12) some errors, e.g. Line 195, “here”; line 254, “blight”; line 420, “chemical deriving”; line 380, “well agrees”; line 339, “contrarily”;
Author Response

(The authors gave the same response as above.)

Reviewer 3 Report
line 123 ... "However,
such texture is not negligible for the quantitative phase fraction analysis as some of the 124
authors previously proposed ".. Please complete the relevant citations.
I recommend larger captions for the picture.
Why you used Gaussian fitting analysis? Not other fitting?
Author Response

(The authors gave the same response as above.)

Round 2
Reviewer 1 Report
The authors addressed all comments and improved the text to the level that can be accepted for publication.
There are minor points still to be adjusted. On line 145, the text from 143 is somehow repeated. The consolidation is needed.
The description of the K vector (line 223) clarifies the values of the x-axis. But in the case of TOF-data is a little bit complicated. In that case, lambda is not constant. The same applies to the description in Chapter 4.1. The authors made here an assumption, using pure Bragg law for constant wavelength instruments, which should be explained. I think this assumption is legitimate for a narrow lambda interval in TOF, but it should be mentioned.
It would be nice also to have the letter description used in Figure 7 in the figure caption and not only in the text.
Author Response
>There are minor points still to be adjusted. On line 145, the text from 143 is somehow repeated. The consolidation is needed.
Repeated descriptions were removed. Thank you.
>The description of the K vector (line 223) clarifies the values of the x-axis. But in the case of TOF-data is a little bit complicated. In that case, lambda is not constant. The same applies to the description in Chapter 4.1. The authors made here an assumption, using pure Bragg law for constant wavelength instruments, which should be explained. I think this assumption is legitimate for a narrow lambda interval in TOF, but it should be mentioned.
As you point out, in our setup, λ is valuable but θ is constant. By using K as the horizontal axis with certain treatments, the diffraction pattern can be understood as like single-wavelength instruments. I think this is widely accepted idea for TOF neutron diffraction experiments.
The original horizontal axis is TOF, but this is a machine-specific value. Every TOF neutron diffractometer has different conversion parameter from TOF to K or d. Therefore, it is necessary to use K to describe the results as the general diffraction phenomenon.
As you pointed out, it is very important to note that the same K among different experiments can be different combinations of λ and θ. The content of section 4.1 is a good example for it. Since Scherrer equation contains λ/cosθ, the results of forward- and back-scattering diffraction experiments should show different sensitivities of the peak broadening.
According to your advice, I added the below explanation at L298. θ in TOF experiment can arbitrarily chosen, basically. However, it is more difficult to distinguish λ of low-angle, relatively fast neutrons by measuring TOF. This is why we had to use the backscattering detectors to see cementite, the minor phase having complex crystal structure.
"Although iMATERIA equips the low 2θ angle detectors as well, the resolution of K becomes worse with decreasing θ [20]. The neutrons corresponding to the K range of interest are too fast to resolve λ by time-of flight measurement."
>It would be nice also to have the letter description used in Figure 7 in the figure caption and not only in the text.
I did so. Thank you very much.
Reviewer 2 Report
I am satisfied that the authors made an effort to implement most of my suggestions. Below, some minor comments for the manuscript before it is going to be published.
1) Line 486, “Wu et al. “, there should be a reference here.
2) Line 490, “carbon densities”.
3) Line 486-488, why in the case of Si-added steel, strain energy is more dominant than solute drag? I suggest the authors either present quantitative calculation result of different contributions of dissipated energies and make more convincing speculation, or voice down the current statement.
4) While Ref. 17 presents a nice and recent review on the effect of alloying elements on the austenite-ferrite phase transformation, some more specific research articles on this topic are recommended to be cited in lines of text on the interaction of alloying elements with the interface (in the Introduction or Discussion sections), such as:
Fang H, van der Zwaag S, van Dijk N H. A novel 3D mixed-mode multigrain model with efficient implementation of solute drag applied to austenite-ferrite phase transformations in Fe-C-Mn alloys. Acta Materialia, 2021, 212: 116897.
Wu H D, Miyamoto G, Yang Z G, et al. Incomplete bainite transformation in Fe-Si-C alloys. Acta Materialia, 2017, 133: 1-9.
Chen H, Zhu K, Zhao L, et al. Analysis of transformation stasis during the isothermal bainitic ferrite formation in Fe–C–Mn and Fe–C–Mn–Si alloys. Acta materialia, 2013, 61(14): 5458-5468.
Author Response
1) Line 486, “Wu et al. “, there should be a reference here.
2) Line 490, “carbon densities”.
They were fixed. Thank you very much for pointing out these.
3) Line 486-488, why in the case of Si-added steel, strain energy is more dominant than solute drag? I suggest the authors either present quantitative calculation result of different contributions of dissipated energies and make more convincing speculation, or voice down the current statement.
Basically, the idea that the strain energy is the dominant part is based on Wu's study, ref [33] (this is the same as one of the papers that you recommend). As you pointed out, our statement at "The formation of strain field around bainitic ferrite corresponds to the above explanation for carbon capturing." was standing on an assumption with weak evidence. Therefore, we removed this sentence.
Maybe we can draw shifted Ae3 by giving additional Gibbs energy for ferrite and fit the experimental data (but no time for this paper, sorry). However, the most important part of this consideration is to find out what is the source of energy dissipation. The current data cannot give any answer for this.
4) While Ref. 17 presents a nice and recent review on the effect of alloying elements on the austenite-ferrite phase transformation, some more specific research articles on this topic are recommended to be cited in lines of text on the interaction of alloying elements with the interface (in the Introduction or Discussion sections), such as:
As mentioned above, we added Wu's study as ref[33]. The papers by Fang and Chen look interesting, too, but I don't have time to read in detail and understood their theories right now. We would like to focus on the energy dissipation theory in the future study. Thank you very much again for your detailed advices and discussion.